# Assessment of Knowledge, Attitude, and Practice towards Tuberculosis among Taif University Students

**DOI:** 10.3390/healthcare11202807

**Published:** 2023-10-23

**Authors:** Eilaf A. Mohammed, Huriyyah A. Alotaibi, Joud F. Alnemari, Meznah S. Althobiti, Shumukh S. Alotaibi, Ashraf A. Ewis, Azza A. K. El-Sheikh, Sayed F. Abdelwahab

**Affiliations:** 1College of Pharmacy, Taif University, Taif 21944, Saudi Arabia; s43800156@students.tu.edu.sa (E.A.M.); s43804056@students.tu.edu.sa (H.A.A.); s43800798@students.tu.edu.sa (J.F.A.); s43810784@students.tu.edu.sa (M.S.A.); S43802037@students.tu.edu.sa (S.S.A.); 2Department of Public Health, Faculty of Health Sciences, Umm Al-Qura University, Makkah 21912, Saudi Arabia; aaewis@uqu.edu.sa; 3Basic Health Sciences Department, College of Medicine, Princess Nourah Bint Abdulrahman University, P.O. Box 84428, Riyadh 11671, Saudi Arabia; aaelsheikh@pnu.edu.sa; 4Department of Pharmaceutics and Industrial Pharmacy, College of Pharmacy, Taif University, Taif 21944, Saudi Arabia

**Keywords:** attitude, knowledge, practice, awareness, pulmonary TB, TU, tuberculosis

## Abstract

Tuberculosis (TB) remains a significant public health concern worldwide. Given the dense living and interactive nature of university environments, students may be at higher risk. This cross-sectional study assessed tuberculosis-related knowledge, attitudes, and practices (KAP) among students at Taif University (TU) from November 2022 to May 2023. Using a self-administered online questionnaire with 40 items, 1155 students participated. Key demographics: 68.2% females, 96.9% Saudi citizens, 94.5% unmarried, and 87.5% non-smokers. Of the respondents, 26.5% had no knowledge of TB. The TB-related KAP scores among the aware students were 64.9%, 74.8%, and 81%, respectively. Medical college students exhibited significantly higher TB-related knowledge and attitudes than their non-medical peers (*p* < 0.001). The findings indicate a commendable level of TB-awareness among TU students, but there remains a substantial uninformed segment. Campaigns to enhance TB knowledge among TU students are suggested.

## 1. Introduction

Tuberculosis (TB) is a bacterial infection that primarily affects the lungs (pulmonary TB), although it can affect other organs in the body [1,2]. It is a reappearing infectious disease and a significant public health problem on a global scale despite advances in diagnosis, treatment, and control [3]. Prior to the corona virus disease-2019 (COVID-19) pandemic, TB was the most common infectious disease that caused mortality [4]. About 90% of those who are infected with TB are adults, and men are more likely than women to be affected [2]. *Mycobacterium tuberculosis* (MTB), an ancestral human pathogen that primarily damages the lungs, is the cause of TB, with pulmonary disease being the most typical manifestation. Despite being mainly a pulmonary infection, TB is a multi-system disease with a broad range of symptoms.

The biggest high-risk age group for contracting TB consists of adolescents between 19 and 22 years, the majority of whom are college and university students [5]. In universities and schools, crowded environments and a high level of person-to-person interactions exist, which makes it easier for diseases, such as TB, to spread [6]. In this environment, a delay in seeking medical help is extremely common. To some extent, failure to diagnose and treat tuberculosis in the initial stage is caused by ignorance about preventive methods. Insufficient levels of knowledge about TB cause ineffective use of medical services and inadequate disease prevention approaches. Consequently, it is critical to describe the degree of existing TB knowledge, attitudes, and practices (KAP) to pinpoint misconceptions and indicate which populations would benefit from interventions to raise TB awareness [5]. Several international studies addressing KAP towards tuberculosis among students at universities have been published. For example, a study was conducted in Bangladesh to assess the knowledge among 839 non-medical university students regarding TB. According to their findings, they concluded that the level of general information about TB is poor among the studied Bangladeshi non-medical students, especially the knowledge regarding latent TB, the DOT program, the curability of TB, and its causative agent as a bacterial infection [7]. Another study addressing TB-related KAP and the choice of communication channels in Thailand found an overall medium level of TB knowledge, with lower levels among migrant and ethnic minority populations and higher levels among family members of TB patients. It was also found that a significant proportion of the population had negative attitudes towards TB patients [8]. Also, a KAP study on tuberculosis in Cameroon showed evidence that Cameroonians’ knowledge of the disease’s symptoms and mode of transmission is frequently incorrect [9]. That study concluded that negative attitudes and poor disease management are impeding the national TB control program in Cameroon [9]. In addition, another study in Rome, Italy, reported that fifth-year medical students have only a moderate understanding of tuberculosis. The internship year students were found to have an increased knowledge of TB diagnosis, epidemiology, and prevention [10]. Moreover, a study in Belgrade, Serbia (*n* = 69 students), showed that TB-related knowledge was inadequate, particularly regarding the cause of TB and the mode of transmission. Most Serbian students wanted to improve their knowledge of tuberculosis [11].

On the other hand, the government of Saudi Arabia (SA) provides free social education. Even though science subjects are covered in basic school education, infectious diseases, such as TB, are not covered in the curriculum except at the medical colleges. In this regard, 3336 TB cases were reported in 2014 [12]. Importantly, the annual incidence of TB in SA was 10/100,000 individuals in 2018 [13] with 41 cases in Taif [14]. Also, the prevalence of TB in Saudia Arabia was about 10.7% over the last decade [15,16]. Therefore, it is necessary for everyone, especially those who play important roles in society, to comprehend TB KAP. The reports that evaluated the TB-related KAP among university students in Saudi Arabia are very limited, as only three reports were found [17,18,19], of which one study was among Taif university (TU) medical students. Assessment of the awareness of university students regarding TB and finding out the gaps in knowledge is very crucial in order to facilitate any upcoming health-educational activities and help the health authorities and policy makers in tailoring efficient and targeted preventive programs for combatting communicable diseases, including TB [20]. Therefore, in the present study, we assessed the KAP of TU students towards TB infection and compared the KAP scores of the students from various colleges (medical, scientific, and humanities) to determine whether any KAP insufficiency might exist and contribute to the elevated TB incidence that could be seen in this setting. 

## 2. Methods

### 2.1. Study Design and Settings

This study was designed as a descriptive cross-sectional study among TU students, and was conducted in Taif City, Saudi Arabia, from November 2022 to May 2023 to assess the TB-related KAP among TU students. No incentive for participation was offered. The study protocol was approved by TU Ethical Committee (Approval # 44-094).

### 2.2. Inclusion/Exclusion Criteria

The study included participants who were students at TU and older than 18 years of age. Those younger than 18 or those from other universities were excluded.

### 2.3. Sample Size Calculation

This study consisted of male and female students in all academic years at TU. A Raosoft Inc., (Seattle, WA, USA) calculator [21] was used to determine the sample size using the Equation *n* = 72 × *p* (1 − *p*)/e^2^ for which *n* is the sample size, z (1.96) is the z-score associated with a level of confidence (95%), p is the sample proportion and is expressed as a decimal, and e (0.05) is the margin of error expressed as a decimal. The calculated sample size was at least 377 participants.

### 2.4. Data Collection and Instrumentation Methods (Data Collection Techniques and Tools)

The study tool was a self-administered questionnaire that was created using the previously available literature. Four sections were included in the questionnaire: Section 1: Socioeconomic questions: the participants were asked about their socio- demographic data to provide information about their gender, age, marital status, nationality, affiliation with different colleges at TU, level of education (academic year) at the university, and other demographics; Section 2: Knowledge assessment questions: this section consisted of 21 questions that measured the students’ general TB knowledge; Section 3: Attitude assessment questions: this section consisted of 11 questions that measured the students’ attitudes towards TB; and Section 4: Practice assessment questions: this section consisted of eight sentences to measure the students’ practices towards tuberculosis.

### 2.5. Calculation of the Knowledge, Attitude, and Practice Scores

For calculation of the knowledge score, each correct answer was given a one-point score, while a wrong answer and an answer of “I do not know” were given a zero-point score. The overall knowledge score was calculated by taking the average score of the participating students who completed the questionnaire. The knowledge score was divided into three categories: (1) poor (<50%), (2) good (50% to <75%), and excellent (≥75%).

Calculation of the attitude score was measured on a 5-point Likert scale. Each answer was given a 1- to 5-point score for which strongly agree was given 5, agree = 4, neutral = 3, disagree = 2, and strongly disagree = 1. Reverse scoring was used when necessary.

The overall attitude score was calculated by taking the average attitude score of the participating students who completed the questionnaire. The attitude score was divided into three categories: (1) poor (<50%), (2) good (50% to <75%), and (3) excellent (≥75%).

Calculation of the practice score was measured on a 5-point Likert scale. Each proper answer was given a 1- to 5-point score for which always was given 5, usually = 4, sometimes = 3, rarely = 2, and never = 1. Reverse scoring was used when necessary. 

The overall practice score was calculated by determining the average practice score of the participating students who completed the questionnaire. The practice score was divided into three categories: (1) poor (<50%), (2) good (50% to <75%), and (3) excellent (≥75%).

### 2.6. Statistical Analysis

Statistical analysis was performed using the statistical package for social science (SPSS) software program version 26 (IBM Inc., Armonk, NY, USA). Descriptive statistics were generated for the responses and presented as frequencies and percentages. For independent variables, the chi-squared test was used to compare categorical variables and their correlation with the sociodemographic data. *p*-Values of <0.05 were considered significant.

## 3. Results

### 3.1. Demographic Characteristics of the Study Participants

This cross-sectional study that used a self-administered online questionnaire was conducted during the period from November 2022 to May 2023. The questionnaire was distributed to students via university email and the available social media platforms. A total of 1244 students at TU took the survey. In total, 89 students (7.2%) did not participate in the study, either because they did not want to participate or because they were younger than 18 years old, while 1155 students (92.8%) agreed and consented to participate in the study. The 1155 participants completed the online questionnaire, and the responses were saved on Google Drive in a password-protected manner. No personal identifiers were collected. The baseline demographic characteristics of the respondents are shown in Table 1. As shown, most of the participants were females (*n* = 788; 68.2%) with ages ranging from 18 to ≥25 years. More than half of the participants (*n* = 626; 54.2%) were 21–24 years old. Also, most of the participants were Saudi citizens (*n* = 1119; 96.9%), and 42.1% (*n* = 486) were students at the medical colleges. The educational level (academic year) with the highest frequency was the third and fourth years (*n* = 477; 41.3%), while the lowest was the fifth, sixth, and internship year students (*n* = 234, 20.3%). Most of the participants were unmarried (*n* = 1091; 94.5%) and non-smokers (*n* = 1011; 87.5%). A high percentage of the participants (*n* = 1095; 94.8%) did not know any people who have/had tuberculosis.

Among the 1155 participants who consented to participate in the study, only 849 heard about TB and finished answering the questionnaire. Participants who had never heard about TB (*n* = 306; 26.4%) were not allowed to proceed with the survey sections of KAP. Consequently, the analysis of KAP was performed with the 849 participants who had heard about TB. The sources of TB knowledge are shown in Table 1.

Students who either heard or did not hear about TB were similar across all demographic characteristics except for age (*p* = 0.001), college, educational level, and knowing people who had TB or not (*p* < 0.001; Table 2). According to Table 2, most of the participants (57.6%) who heard about TB belonged to the age group of 21 to 24 years. A high percentage of younger age students had not heard about TB. Also, participants from the first and second academic years were less likely to have heard about TB. In addition, half of the participants (50.5%) who heard about TB were from medical colleges. Interestingly, students who knew people who had TB were more likely to have heard about it (*p* = 0.001; odds ratio 16.7 and confidence interval = 2.38–116.9; Table 2).

### 3.2. Assessment of the Knowledge of the Study Participants about TB

The knowledge questions were scored as one point for the correct answer and zero for the incorrect answer (Table 3). Less than half of the respondents knew that the cause of TB is a bacterium (44.3% of the total participants), and 76.3% of them correctly answered the most affected organ by TB. Also, 57% of the participants knew that a vaccine for TB was available, and about one-third of them (33.1%) did not know about such availability. In addition, most of the participants (70.6%) answered yes to “TB can be cured”. Moreover, 74.8% of them knew that TB can be transmitted by air when a person with TB coughs or sneezes, and only 21.4% chose consumption of raw unpasteurized milk as a source of infection. By contrast, most of the participants did not know the correct duration of TB treatment. The answers to the remaining knowledge questions are shown in Table 3.

### 3.3. Assessment of the Attitude of the Study Participants towards TB

Table 4 shows the respondents’ attitudes towards TB. Among the 849 participants, most respondents had a high level of awareness about TB. About 85.7% strongly agreed that “if they develop tuberculosis, they must immediately notify their family and/or physician”. Also, 80.8% of the participants strongly agreed that they would encourage those around them who have tuberculosis to seek treatment. In addition, 79.6% strongly agreed that “If they had TB, they would take anti-TB medications on regular basis, as prescribed by their doctor”. Results in the attitude sections reflect a proper attitude towards TB among TU students with a good/excellent attitude score of 74.8% (details described below).

### 3.4. Assessment of the Practice of the Study Participants towards TB

Eight questions assessed the practices related to TB infection among the participants, which are shown in Table 5. As shown, 79.2% answered that they always cover their mouths when they sneeze or cough for fear of spreading infections. Also, 71.6% of the participants answered that they always open the window to let fresh air enter with the aim of preventing infections. In addition, 68.2% of the participants answered that they always wash their hands or use hand sanitizer after going to the hospital. Most of the participants answered that they always follow the healthiest practices, such as wearing a face mask in infective environment, maintaining a healthy diet, and exercising regularly (Table 5).

### 3.5. Assessment of Knowledge Score Regarding TB

The knowledge score was calculated and divided into three categories as described in the Subjects and Methods section. According to Table 6, the good/excellent knowledge score of all the 849 participants was found to be 64.9%. Most of the participants had a good/excellent TB knowledge score that correlated with many demographic characteristics. In this regard, a good/excellent knowledge score was high in the age group 21–24 (68.1%; <0.001) when compared to other age groups. Also, participants from medical colleges had the highest percentage of a good/excellent knowledge score (*p* < 0.001) when compared to other college groups. In addition, participants from the first and second academic years were less likely to have a good/excellent knowledge score regarding TB when compared with students in the higher academic levels (*p* < 0.001). Furthermore, the students who knew people with TB infection were more likely to have a higher knowledge score as compared with those who did not (*p* = 0.02). We did not find any significant correlation between knowledge scores and participants’ genders, nationalities, marital and smoking states (Table 6).

### 3.6. Assessment of Attitude Score Regarding TB

The attitude scores of the study participants were calculated and classified as described in the Subjects and Methods section. According to Table 7, most participants obtained good/excellent attitude scores towards TB without any significant correlation with any demographic characteristics except the medical colleges’ students, who obtained higher percentage of good/excellent attitude scores (80.5%) when compared with scientific (68.5%) and humanities college students (69.6%; *p* < 0.001; Table 7).

### 3.7. Assessment of Practice Scores Regarding TB

The practice scores of the study participants were calculated and classified as described in the Subjects and Methods section. In Table 8, a good/excellent practice score was recorded among 81% of the 849 participants. Most of the participants (*n* = 506; 59.6%) obtained a good TB-related practice score. A significant difference in the practice score based on marital status (*p* = 0.031) was found for which married students had a significantly higher proportions of good/excellent practice score as compared to their peers. Also, if the participants knew someone who had TB infection, they were more likely to have a good practice score when compared with their peers who did not (*p* = 0.016). In this regard, the frequency of participants who knew people who had TB and obtained a good/excellent practice score (94.9%) was higher than those who did not (80%; Table 8).

## 4. Discussion

This study was conducted to assess TB-related KAP among students at different colleges of TU. A total of 1155 individuals participated in the study with only 849 (73.5%) students hearing about TB whereas 26.5% of them had never heard about it. The level of TB-related knowledge among the 849 participants was good. The attitudes and practices of the participants were aimed at avoiding TB infection. Students at the medical colleges were more likely to have a good/excellent TB-related knowledge and attitudes (*p* < 0.001 each) when compared with their peers in non-medical colleges. Several outcomes of this analysis deserve further discussion.

First, almost no data in the literature have examined the TB-related KAP among TU students. A recent literature search found only one study that examined KAP regarding TB among 435 students at only the TU medical college [17], whereas our study included students from medical, scientific, and humanities colleges. Also, the previous study [17] examined only the knowledge of the participants and did not examine the attitudes and practices. Most of our 849 participants obtained a good/excellent knowledge score about TB, which correlated with many participants’ demographic characteristics, such as the age of the participants (*p* < 0.001). A study conducted at Unaizah College of Medicine, Saudi Arabia, showed that the students whose age was 23 years old had higher levels of TB knowledge [18]. That study [18] targeted only medical students and found that medical students have inadequate knowledge and attitudes towards TB, particularly among those in the early academic levels. The findings of the study conducted at King Saud University [19] are consistent with our results as they showed that the participants who study or work in the healthcare sector showed higher levels of awareness. In addition, the results regarding academic levels align with our results, as we found that participants from the first two academic years were less likely to have good knowledge when compared to higher academic levels (*p*< 0.001). Other reports from Bhutan and Bangladesh [22,23] also concur with our findings. This finding might be attributed to the fact that young people are open to receiving information. Two other studies [22,24] support our results, showing that those with higher academic levels and postgraduate students tend to have better TB-related knowledge of TB.

Second, about 44.3% of our participants knew that a bacterium causes TB, which is a low percentage when compared with a study among social media users in Bangladesh [25]. The latter study found that 71% of the participants knew that germs or bacteria are the primary cause of TB. Our participants’ knowledge of TB transmission was above average (74.8%). In this regard, a study of TB-related KAP in Surulere, Lagos, Nigeria [26] found that only 48.0% of the participants correctly stated that TB is an airborne infection and slightly more than half (52.4%) of them identified overcrowding as aiding TB transmission, a finding that coincides with our findings. In our study, awareness of TB vaccination was intermediate, with 57% of the participants aware of a vaccine against TB. In a study regarding community knowledge and attitudes towards pulmonary TB in rural Edo State, Nigeria [27], 47.3% of the respondents knew about the Bacillus Calmette–Guérin (BCG) vaccination. Regarding the curability of TB, 70.6% of our participants believed it is curable. Similarly, a cross-sectional study [28] found that 75.7% of the participants knew that TB is utterly curable, whereas 24.3% still assumed that TB was a non-curable disease. On the other hand, the gender variable did not yield any significance in our result. However, compared to other studies, women exhibited better knowledge of TB than men [6,29]. Also, we did not find any significant difference regarding marital status and knowledge score in our study. In comparison, a report from Indonesia found that married people were more likely to have higher knowledge than those who were single. The last report points to the likelihood that married individuals have more opportunities to discuss TB with their partners [30]. To this end, it is important to determine whether knowledge of the participants is reflected in their attitudes and practices.

Third, our participants’ attitudes towards TB significantly correlated with the college variable (*p* < 0.001). As expected, medical colleges had a higher attitude score than other colleges (Science and Humanities). A study conducted in Sweden found that non-healthcare students have a negative attitude towards TB [31]. Two other studies [10,32] also concurred with our results in that the medical students have a positive attitude towards TB. Similarly, a recent study [33] found a significant difference concerning the levels of KAP regarding TB between the different colleges of the participants. This difference could be due to the formal teaching of TB by medical colleges in the undergraduate courses, which are integrated with each health college’s internship curriculum. Students from health colleges receive formal training on health topics, particularly TB, in the field of infectious diseases. As a result, students in the health colleges are better informed, more aware, and have positive attitudes and practices towards preventing contagious diseases, such as TB. Other factors could also affect a person’s attitude. A study conducted in Lesotho [30] found that female adults aged 33–44 and those exposed to media were more likely to have a positive attitude towards TB. When it comes to the general population outside of universities, in a study in the Makkah region, Saudi Arabia, residents demonstrated that 89.9% of the respondents demonstrated poor attitudes, whereas only 2.3% had good attitudes [29]. Another study assessed KAP in the Saudi general population and showed that the attitude of the respondents about TB was negative among most of the participants [34]. In our study, most of the participants had a good attitude score. Similarly, a study in eastern Ethiopia [35] found that more than half of the participants considered TB a severe disease. The proper attitudes and practices (please see below) reported in this paper may be attributed to the recent COVID-19 pandemic and the increased awareness of transmission and prevention of infectious diseases in general.

Fourth, our participants’ good/excellent practice scores were high (81%). Participants in our study correctly believed that it was wise to “cover their mouths when they sneeze” and 79.2% of them chose the “always” answer to that question. These responses were significantly associated with the age and marital status of the participants (*p* = 0.001). On the other hand, the percentage of participants who chose “always” in response to “Open the window to let fresh air enter to avoid infection” was 71.6%, while “never” and “rarely” were only 3.5%. This result shows positive practices among the participants towards TB. Also, in our study, only 3.2% of the participants chose “never” concerning “wearing a facemask when visiting a hospital.” This practice is considered good because TB is mainly transmitted in an airborne manner from one person to another. Marital status and knowing people who have/had TB were significantly associated with the respondent’s practice towards TB. Regarding marital status, poor practice scores were higher among unmarried participants, indicating that married people have more awareness regarding TB infection and ways which prevent it. The potential explanation is that married people are responsible for their children, which makes them fearful of infectious disease and willing to be aware of anything harmful to their kids, especially regarding their health. In contrast, a recent study conducted in Bangladesh [25] found that married people showed poor practices towards TB. The difference could be attributed to the differences in the characteristics of the specific populations. As for hygienic practices, a recent study among Jordanian university students [36] found that most participants indicated that they sometimes use hygiene products in public (43.9%), while others always use them (40.0%), and only 16.1% do not use them at all. Notably, a large proportion of participants either were not sure (25.1%) or did not believe (15.9%) that masks can prevent the transmission of airborne infections.

Fifth, we found that the TB-related knowledge scores of the students at the medical colleges were higher than those of students in the humanities and scientific colleges. This difference could be attributed to the nature of the courses being taught at these colleges. However, the attitude and practice scores of the students in different college groups were not significantly different. This similarity can be attributed to the recent increase in awareness by most of the population of the COVID-19 pandemic that started at the beginning of 2020. This was reflected in a high attitude and practice scores among all the study participants. To raise the TB-related knowledge among non-medical students, we suggest that the university management plans awareness campaigns, seminars and workshops during the orientation at the beginning of each academic year.

Finally, although this study has several strengths, including the large number of participants and provision of important information regarding KAP towards TB, it also has several limitations. These limitations include questionnaire-based limitations in the data collection, such as refusal to participate in the study, which is expected to happen. Also, the study showed a female dominance in participation (68.2%), which may be attributed to the frequency of social media usage by females. In addition, this study was conducted as a cross-sectional study. Therefore, the research results may not apply to other settings.

## 5. Conclusions

In conclusion, the level of TB-related knowledge among the 849 participants was good, while nearly 26% of the participants had not heard about TB. The attitudes and practices of the participants were aimed at avoiding TB infection. Despite the gaps in TB-related knowledge, the participants generally showed positive attitudes and obtained positive practice scores. Awareness campaigns and workshops are required to increase the knowledge of TU students about TB, particularly among humanities and scientific college students.

## Figures and Tables

**Table 1 healthcare-11-02807-t001:** Baseline demographic characteristics of the respondents (*n* = 1155).

Variable	Subcategory	Frequency (Percent)
Gender	Male	367 (31.8)
	Female	788 (68.2)
Age (years)	18–20	458 (39.7)
	21–24	626 (54.2)
	≥25	71 (6.1)
Nationality	Saudi	1119 (96.9)
	Non-Saudi	36 (3.1)
Type of College	Medical Colleges ^#^	486 (42.1)
	Science and Engineering Colleges ^#^	320 (27.7)
	Humanities, Education, Sharia and Administrative Colleges ^#^	349 (30.2)
Educational Level	First and Second Year	444 (38.4)
	Third and Fourth Year	477 (41.3)
	Fifth, Sixth, and Internship Year	234 (20.3)
Marital Status	Single	1091 (94.5)
	Married	60 (5.2)
	Divorced/Widowed	4 (.3)
Smoking	Yes	110 (9.5)
	No	1011 (87.6)
	Ex-smoker	34 (2.9)
Do you know people who have/had tuberculosis (TB)?	Yes	60 (5.2)
	No	1095 (94.8)
Have you heard about TB?	Yes	849 (73.5)
	No	306 (26.5)
From where have you heard about TB?	Newspapers, Radio, TV Show, or Social Media.	230 (27.1)
	Newspapers, Radio, TV Show, Social Media, Family, Friends, Neighbors, Healthcare Workers, or Colleagues.	75 (8.8)
	Newspapers, Radio, TV Show, or Social Media, Family, Friends, Neighbors, Healthcare Workers, Colleagues, Brochures, Posters, Lectures, or Other Printed Materials.	100 (11.8)
	Newspapers, Radio, TV Show, Social Media, Brochures, Posters, Lectures, or Other Printed Materials.	96 (11.3)
	Family, Friends, Neighbors, Healthcare Workers, or Colleagues.	123 (14.5)
	Family, Friends, Neighbors, Healthcare Workers, Colleagues, Brochures, Posters, Lectures, or Other Printed Materials.	45 (5.3)
	Brochures, Posters, Lectures, or Other Printed Materials.	180 (21.2)

^#^ Medical Colleges include Medicine, Dentistry, Pharmacy, and Applied Medical Sciences. Science and Engineering Colleges include Science, Engineering, Computer Science and Information Technology, Design and Applied Arts. Humanities, Education, Sharia and Administrative Colleges include Sharia and Regulations, Business Administration, Literature, Education, and Applied College.

**Table 2 healthcare-11-02807-t002:** Comparison of sociodemographic characteristics of participants according to hearing about TB.

Variable	Know about TB *n* = 849 (73.5%)	Do Not Know about TB, *n* = 306 (26.5%)	*p*	Odds Ratio Confidence Interval
**Gender**			0.912	
Male	269 (31.7)	98 (32.0)		1
Female	580 (68.3)	208 (68.0)		1.01 (0.93–1.08)
**Age (years)**			**<0.001**	
18–20	311 (36.6)	147 (48.0)		1
21–24	489 (57.6)	137 (44.8)		1.87 (1.08–1.94)
≥25	49 (5.8)	22 (7.2)		1.04 (0.71–1.50)
**Nationality**			0.345	
Saudi	825 (97.2)	294 (96.1)		1
Non-Saudi	24 (2.8)	12 (3.9)		1.11 (0.87–1.39)
**Type of College**			**<0.001**	
Medical Colleges ^#^	429 (50.5)	57 (18.6)		2.39(1.23–5.09)
Science and Engineering Colleges ^#^	213 (25.1)	107 (35.0)		1.08 (0.32–1.27)
Humanities and Education ^#^	207 (24.4)	142 (46.4)		1
**Educational level**			**<0.001**	
First and Second Year	295 (34.8)	149 (48.7)		1
Third and Fourth Year	368 (43.3)	109 (35.6)		1.97 (1.01–2.82)
Fifth, Sixth and Internship Year	186 (21.9)	48 (15.7)		0.86 (0.74–1.24)
**Marital status**			0.319	
Single	804 (94.7)	287 (93.8)		1
Married	41 (4.8)	19 (6.2)		1.08 (0.91–1.29)
Divorced/Widowed	4 (0.5)	0 (0.0)		0.74 (0.71–0.78)
**Smoking**			0.319	
Yes	75 (8.8)	35 (11.4)		1
No	747 (88.0)	264 (86.3)		0.98 (0.88–1.18)
Ex-smoker	27 (3.2)	7 (2.3)		1.11 (0.89–1.46)
**Do you know people who have/had TB?**			**<0.001**	
Yes	59 (6.9)	1 (0.3)		16.7 (2.38–116.9)
No	790 (93.1)	305 (99.7)		1

^#^ Medical Colleges include Medicine, Dentistry, Pharmacy, and Applied Medical Sciences. Science and Engineering Colleges include Science, Engineering, Computer Science and Information Technology, Design and Applied Arts. Humanities, Education, Sharia and Administrative Colleges include Sharia and Regulations, Business Administration, Literature, Education, and Applied Colleges. Bold numbers indicate statistical significance.

**Table 3 healthcare-11-02807-t003:** Knowledge of the study participants (*n* = 849) by frequency among those who had heard about TB.

Knowledge Questions			Frequency (%)
**TB is caused by:**			
-Bacterium			376 (44.3)
-Parasite			12 (1.4)
-Fungus			3 (0.4)
-Virus			251 (29.6)
-I don’t know			207 (24.4)
**The organ most affected by TB is:**			
-Skin			5 (6.0)
-GIT			10 (1.2)
-Lung			648 (76.3)
-Heart			6 (0.7)
-Liver			54 (6.4)
-Kidney			15 (1.8)
-I don’t know			111 (13.1)
**Questions**	**Yes,** ***n* (%)**	**No,** ***n* (%)**	**I don’t know,** ***n* (%)**
-Is there is a vaccine for TB?	484 (57.0)	84 (9.9)	281 (33.1)
-TB can be cured?	599 (70.6)	74 (7.7)	176 (20.7)
-Drug resistance develops in the patient on discontinuation of treatment	407 (47.9)	84 (9.9)	358 (42.2)
-If a person has a TB patient among his family or friends, he should be tested for the disease	599 (70.5)	77 (9.1)	173 (20.4)
-The TB microorganism circulates in the air	416 (49.0)	188 (22.1)	245 (28.9)
-A single dose of the BCG vaccine can provide lifetime protection against tuberculosis infection	195 (23.0)	182 (21.4)	472 (55.6)
-Is TB a contagious disease?	567 (66.8)	109 (12.8)	173 (20.4)
-TB vaccination is taken at the age of 6 months	345 (40.6)	66 (7.8)	438 (51.6)
**Which of the following are TB symptoms?**			
-Fever	675 (79.5)	32 (3.8)	142 (16.7)
-Constipation	133 (15.7)	270 (31.8)	446 (52.5)
-Weight loss	483 (56.9)	76 (9.0)	290 (34.2)
-Cough or cough with blood	748 (88.1)	13 (1.5)	88 (10.4)
-Night sweat	510 (60.1)	38 (4.5)	301 (35.5)
-Muscle pain	413 (48.6)	77 (9.1)	359 (42.3)
**Risk Factors of TB infection including the following:**			
-Smoking	618 (72.8)	49. (5.8)	182 (21.4)
-Heart disease	321 (37.8)	173 (20.4)	355 (41.8)
-Malnutrition	394 (46.4)	142 (16.7)	313 (36.9)
-Physical inactivity	246 (29.0)	208 (24.5)	395 (46.5)
-HIV infection	596 (70.2)	40 (4.7)	213 (25.1)
-Diabetes mellitus	279 (32.9)	154 (18.1)	416 (49.0)
**TB could be transmitted through:**			
-Handshakes with TB patients	291 (34.3)	301 (35.4)	257 (30.3)
-The air when a person with TB coughs or sneeze	635 (74.8)	73 (8.6)	141 (16.6)
-Touching items in public places that have been touched or used by TB patients	403 (47.5)	188 (22.1)	258 (30.4)
-Sexual transmission of an infection from an infected individual to their partner	363 (42.8)	180 (21.2)	306 (36.0)
-Consuming uncooked milk	182 (21.5)	216 (25.4)	451 (53.1)
-Contaminated blood transfusions	524 (61.7)	79 (9.3)	246 (29.0)
**Which of the following treatments is most effective for treating TB?**			
-Herbal medicine/traditional healers	118 (13.9)	424 (49.9)	307 (36.2)
-Specific medicine	705 (83.0)	23 (2.7)	121 (14.3)
-Buy OTC drug	61 (7.2)	614 (72.3)	174 (20.5)
**The treatment duration of TB is:**			
1–2 weeks	80 (9.4)	365 (43.0)	404 (47.6)
1–2 months	141 (16.6)	302 (35.6)	406 (47.8)
6–9 months	398 (46.9)	64 (7.5)	387 (45.6)
**Which of the following are TB prevention methods?**			
-Avoiding TB patients	580 (68.3)	105 (12.4)	164 (19.3)
-By maintaining a healthy diet and engaging in a lot of physical activity	515 (60.7)	135 (15.9)	199 (23.4)
-Maintaining a normal blood pressure	396 (46.6)	181 (21.3)	272 (32.0)
-By wearing a mask while interacting with someone who has an Infection	694 (81.7)	37 (4.4)	118 (13.9)
-By residing in ventilated homes	666 (78.5)	41 (4.8)	142 (16.7)

**Table 4 healthcare-11-02807-t004:** Attitudes of study participants (*n* = 849) by frequency among those who have heard about TB.

Attitude Question	Responses N (%)				
	Strongly Agree (SA)	Agree (A)	Neutral (N)	Disagree (D)	Strongly Disagree (SD)
If I develop tuberculosis, I must immediately notify my family and/or Physician.	728 (85.7)	63 (7.4)	39 (4.7)	8 (0.9)	11 (1.3)
If I had TB, it might not decrease the quality of my life and daily tasks.	140 (16.5)	144 (17.0)	230 (27.1)	241 (28.3)	94 (11.1)
Tuberculosis screening is difficult to access at health facilities such as community health centers.	150 (17.7)	138 (16.3)	250 (29.4)	188 (22.1)	123 (14.5)
I would encourage those around me who have tuberculosis to seek treatment.	686 (80.8)	77 (9.1)	59 (6.9)	17 (2.0)	10 (1.2)
TB is regarded as a serious disease.	421 (49.6)	227 (26.7)	160 (18.8)	26 (3.1)	15 (1.8)
Annual medical examinations can’t help prevent tuberculosis.	94 (11.1)	70 (8.2)	207 (24.4)	250 (29.4)	228 (26.9)
Education regarding TB is desperately needed.	91 (10.7)	44 (5.2)	92 (10.8)	218(25.7)	404 (47.6)
Quarantine of tuberculosis patients may reduce the risk of TB spread.	406 (47.8)	157 (18.5)	219 (25.8)	45 (5.3)	22 (2.6)
The public has significant role in controlling TB.	505 (59.5)	179 (21.1)	137 (16.1)	17 (2.0)	11 (1.3)
If I had TB, the inability to leave work or lack of knowledge where to go is the reason of not visiting a healthcare facility.	142 (16.7)	140 (16.5)	304 (35.8)	146 (17.2)	117 (13.8)
If I had TB, I would take anti-TB medications on regular bases, as prescribed by my doctor.	676 (79.6)	80 (9.4)	71 (8.4)	11 (1.3)	11 (1.3)

**Table 5 healthcare-11-02807-t005:** Practice of the study participants (*n* = 849) by frequency among those who have heard about TB.

Practice Question	Responses N (%)				
	Always *n* (%)	Usually *n* (%)	Sometimes *n* (%)	Rarely *n* (%)	Never *n* (%)
I cover my mouth when I sneeze or cough for fear of spreading infections.	672 (79.2)	109 (12.8)	56 (6.6)	8 (0.9)	4 (0.5)
I open the window to let fresh air enter to avoid infections.	608 (71.6)	123 (14.5)	88 (10.4)	22 (2.6)	8 (0.9)
I eat a well-balanced and nutritious diet to stay healthy to avoid infections.	446 (52.5)	149 (17.6)	188 (22.1)	49 (5.8)	17 (2.0)
I wash my hands or use a hand sanitizer after going to the hospital.	579 (68.2)	133 (15.6)	93 (11.0)	31(3.7)	13 (1.5)
I exercise on regular bases to maintain my health.	398 (46.9)	156 (18.3)	186 (21.9)	76 (9.0)	33 (3.9)
I wear facemasks when visiting a hospital.	475 (55.9)	137 (16.1)	145 (17.1)	65 (7.7)	27 (3.2)
I would visit a health facility, if my cough persisted for longer than two weeks to get examined.	541 (63.7)	144 (17.0)	120 (14.1)	28 (3.3)	16 (1.9)
I read materials intended to raise awareness about TB.	419 (49.4)	127 (15.0)	167 (19.6)	88 (10.4)	48 (5.6)

**Table 6 healthcare-11-02807-t006:** Assessment of the knowledge score of the study participants towards TB.

Variable	Knowledge Score N (%)			
	Excellent *n* = 165 (19.4%)	Good *n* = 386 (45.5%)	Poor *n* = 298 (35.1%)	*p*-Value
**Gender**				0.142
Male	53 (19.7%)	110 (40.9%)	106 (39.4%)	
Female	112 (19.3%)	276 (47.6%)	192 (33.1%)	
**Age (years)**				**<0.001**
18–20	32 (10.3%)	153 (49.2%)	126 (40.5%)	
21–24	125 (25.6%)	208 (42.5%)	156 (31.9%)	
≥25	8 (16.3)	25 (51.0%)	16 (32.7%)	
**Nationality**				0.292
Saudi	161 (19.5%)	378 (45.8%)	286 (34.7%)	
Non-Saudi	4 (16.7%)	8 (33.3%)	12 (50.0%)	
**Type of College**				**<0.001**
Medical Colleges ^#^	134 (31.3%)	197 (45.9%)	98 (22.8%)	
Science and Engineering Colleges ^#^	17 (8.0%)	97 (45.5%)	99 (46.5%)	
Humanities, Education, etc. ^#^	14 (6.8%)	92 (44.4%)	101 (48.8%)	
**Educational level**				**<0.001**
1st and 2nd	29 (9.8%)	141 (47.8%)	125 (42.4%)	
3rd and 4th	72 (19.5%)	168 (45.7%)	128 (34.8%)	
5th and 6th and above	64 (34.4%)	77 (41.4%)	45 (34.2%)	
**Marital status**				0.916
Single	157 (19.5%)	366 (45.5%)	281 (35.0%)	
Married +	8 (17.8%)	20 (44.4%)	17 (37.8%)	
**Smoking**				0.245
Non-smoker	151 (20.2%)	339 (45.4%)	257 (34.4%)	
Smoker *	14 (13.7%)	47 (46.1%)	41 (40.2%)	
**Do you know people who have/had TB?**				**0.02**
Yes	9 (15.3%)	37 (62.7%)	13 (22.0%)	
No	156 (19.7%)	349 (44.2%)	285 (36.1%)	

^#^ Medical Colleges include Medicine, Dentistry, Pharmacy, and Applied Medical Sciences. Science and Engineering Colleges include Science, Engineering, Computer Science and Information Technology, Design and Applied Arts. Humanities, Education, Sharia and Administrative Colleges include Sharia and Regulations, Business Administration, Literature, Education, and Applied Colleges. Bold numbers indicate statistical significance. + is married plus Divorced/Widow, * is smokers and ex-smokers.

**Table 7 healthcare-11-02807-t007:** Assessment of the attitude score of the study participants towards TB.

Variable	Attitude Score N (%)			
	Excellent *n* = 459 (54.1%)	Good *n* = 176 (20.70%)	Poor *n* = 214 (25.20%)	*p*-Value
**Gender**				0.256
Male	149 (55.4%)	47 (17.5%)	73 (27.1%)	
Female	310 (53.4%)	129 (22.3%)	141 (24.3%)	
**Age (years)**				0.244
18–20	166 (53.4%)	63 (20.2%)	82 (26.4%)	
21–24	273 (55.8%)	102 (20.9%)	114 (23.3%)	
≥25	20 (40.8%)	11 (22.4%)	18 (36.8%)	
**Nationality**				0.595
Saudi	444 (53.8%)	171 (20.7%)	210 (25.5%)	
Non-Saudi	15 (62.5%)	5 (20.8%)	4 (16.7%)	
**Type of College**				**<0.001**
Medical Colleges ^#^	267 (62.2%)	78 (18.3%)	84 (19.5%)	
Science and Engineering Colleges ^#^	97 (45.5%)	49 (23.0%)	67 (31.5%)	
Humanities, Education, etc. ^#^	95 (45.9%)	49 (23.7%)	63 (30.4%)	
**Educational level**				0.667
1st and 2nd	150 (50.8%)	64 (21.7%)	81 (27.5%)	
3rd and 4th	202 (54.9%)	75 (20.4%)	91 (24.7%)	
5th and 6th and above	107 (57.5%)	37 (19.9%)	42 (22.6%)	
**Marital status**				0.13
Single	438 (54.5%)	169 (21.0%)	197 (24.5%)	
Married +	21 (46.7%)	7 (15.5%)	17 (37.8%)	
**Smoking**				0.29
Non-smoker	411 (55.0%)	153 (20.5%)	183 (24.5%)	
Smoker *	48 (47.1%)	23 (22.5%)	31 (30.4%)	
**Do you know people who have/had TB?**				0.156
Yes	38 (64.4%)	7 (11.9%)	14 (23.7%)	
No	421 (53.3%)	169 (21.4%)	200 (25.3%)	

^#^ Medical Colleges include Medicine, Dentistry, Pharmacy, and Applied Medical Sciences. Science and Engineering Colleges include Science, Engineering, Computer Science and Information Technology, Design and Applied Arts. Humanities, Education, Sharia and Administrative Colleges include Sharia and Regulations, Business Administration, Literature, Education, and Applied Colleges. Bold numbers indicate statistical significance. + is married plus Divorced/Widow, * is smokers and ex-smokers.

**Table 8 healthcare-11-02807-t008:** Assessment of the practice score of the study participants towards TB.

Variable	Practice Score N (%)			
	Excellent *n* = 506 (59.6%)	Good *n* = 182 (21.4%)	Poor *n* = 161 (19%)	*p*-Value
**Gender**				0.387
Male	166 (61.7%)	50 (18.6%)	53 (19.7%)	
Female	340 (58.6%)	132 (22.8%)	108 (18.6%)	
**Age (years)**				0.725
18–20	183 (58.8%)	68 (21.9%)	60 (19.3%)	
21–24	293 (59.9%)	101 (20.7%)	95 (19.4%)	
≥25	30 (61.2%)	13 (26.5%)	6 (12.3%)	
**Nationality**				0.764
Saudi	490 (59.4%)	178 (21.6%)	157 (19%)	
Non-Saudi	16 (66.7%)	4 (16.7%)	4 (16.7%)	
**Type of College**				0.363
Medical Colleges ^#^	260 (60.6%)	98 (22.8%)	71 (16.6%)	
Science and Engineering Colleges ^#^	129 (60.5%)	40 (18.8%)	44 (20.7%)	
Humanities, Education, etc. ^#^	117 (56.5%)	44 (21.3%)	46 (22.2%)	
**Educational level**				0.187
1st and 2nd	174 (59.0%)	71 (24.1%)	50 (16.9%)	
3rd and 4th	228 (62.0%)	65 (17.6%)	75 (20.4%)	
5th, 6th and above	104 (55.9%)	46 (24.7%)	36 (19.4%)	
**Marital status**				**0.031**
Single	479 (59.6%)	167 (20.7%)	158 (19.7%)	
Married +	27 (60.0%)	15 (33.3%)	3 (6.7%)	
**Smoking,**				0.339
Non-smoker	452 (60.5%)	156 (20.9%)	139 (18.6%)	
Smoker *	54 (52.9%)	26 (25.5%)	22 (21.6%)	
**Do you know people who have/had TB?**				**0.016**
Yes	43 (72.9%)	13 (22.0%)	3 (5.1%)	
No	463 (58.6%)	169 (21.4%)	158 (20.0%)	

^#^ Medical Colleges include Medicine, Dentistry, Pharmacy, and Applied Medical Sciences. Science and Engineering Colleges include Science, Engineering, Computer Science and Information Technology, Design and Applied Arts. Humanities, Education, Sharia and Administrative Colleges include Sharia and Regulations, Business Administration, Literature, Education, and Applied Colleges. Bold numbers indicate statistical significance. + is married plus Divorced/Widow, * is smokers and ex-smokers.

## Data Availability

All data associated with this study are included herein.

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
