# Peer review of "Assessment of Knowledge, Attitude, and Practice towards Tuberculosis among Taif University Students"

_healthcare, 2023, doi:10.3390/healthcare11202807_

Round 1
Reviewer 1 Report
The work is good and relevant in a public health context. However, minor revisions are necessary:
- Abstract and remain manuscript: ilness vs disease? the word should be harmonised throughout the manuscript. I suggest keeping disease.
- Abstract: "Students are at a significant risk of getting TB due to the overcrowded environment" - this sentence is not an absolute truth. It only happens in areas where TB is a common/endemic disease. It is necessary to contextualise Saudi Arabia in terms of the prevalence of TB in this country (in the introduction!). The sentence should be rephrased to be true only in areas at risk for TB.
- Introduction: It has too much information from other equivalent studies. This information is repeated in the discussion. It should be reduced to the specific information of the equivalent studies in the introduction and be highlighted in the discussion of the manuscript. Please rephrase the second paragraph of the introduction!
- Last paragraph of introduction: "In our study, a total of 1155 filled out the online questionnaire and 306 of them did not hear about TB. The knowledge score of the remaining 849 students about TB was 64.9% and their attitudes and practices were towards avoiding TB infection." - This is results! The introduction should end with the objectives of the study (these need to be clarified!) and the paragraph should be in the M&M and results chapter. Reformulate!
- Discussion and conclusions: there should be more discussion about the next steps to reverse this lack of knowledge among the students. Scientific communication... training actions for students... etc etc
English language of the manuscript: please revise and harmonise some expressions that are the same but the authors use differents words to the same expression. e.g. ilness vs disease
Author Response
Response to "REVIEWER #1"
General comment: The work is good and relevant in a public health context. However, minor revisions are necessary:
Response to the General comment: We thank the reviewer for commending the manuscript.
Comment #1: Abstract and remain manuscript: illness vs disease? the word should be harmonized throughout the manuscript. I suggest keeping disease.
Response to Comment #1: We thank the reviewer for the comment; and the manuscript was revised to reflect the reviewer’s suggestion (“illness” was replaced by “disease” throughout the manuscript).
Comment #2: Abstract: "Students are at a significant risk of getting TB due to the overcrowded environment" - this sentence is not an absolute truth. It only happens in areas where TB is a common/endemic disease. It is necessary to contextualize Saudi Arabia in terms of the prevalence of TB in this country (in the introduction!). The sentence should be rephrased to be true only in areas at risk for TB.
Response to Comment #2: We thank the reviewer for the comment; and the abstract was revised to reflect the reviewer’s suggestion. Also, the introduction was revised to include the prevalence and incidence of TB in Saudi Arabia (Please find the changes in the last paragraph of the introduction).
Comment #3: Introduction: It has too much information from other equivalent studies. This information is repeated in the discussion. It should be reduced to the specific information of the equivalent studies in the introduction and be highlighted in the discussion of the manuscript. Please rephrase the second paragraph of the introduction!
Response to Comment #3: The introduction was revised to reflect the reviewer’s point of view. The study in Thailand was moved after that in Bangladesh (Asia), followed by the Cameroonian study, and the two studies in Europe were consecutive to each other. This paragraph was rearranged and shortened.
Comment #4: Last paragraph of introduction: "In our study, a total of 1155 filled out the online questionnaire and 306 of them did not hear about TB. The knowledge score of the remaining 849 students about TB was 64.9% and their attitudes and practices were towards avoiding TB infection." This is results! The introduction should end with the objectives of the study (these need to be clarified!) and the paragraph should be in the M&M and results chapter. Reformulate!
Response to Comment #4: We thank the reviewer for the comment. However, best practice scientific writing usually includes an executive summary of the results in the last paragraph of the introduction. Therefore, we prefer to leave this part as is.
Comment #5: Discussion and conclusions: there should be more discussion about the next steps to reverse this lack of knowledge among the students. Scientific communication... training actions for students... etc.
Response to Comment #5: We thank the reviewer for the comment and the discussion was modified to reflect the reviewer’s suggestion (Please, check this modification in the “fifth” section of the Discussion).
Comment #6: English language of the manuscript: please revise and harmonize some expressions that are the same but the authors use different words to the same expression. e.g. illness vs disease
Response to Comment #6: The manuscript was edited by a native English speaker. An editing certificate is submitted along with the revision.
Reviewer 2 Report
Reviewer Comments
September 7, 2023
Abstract Section:
The description that students are prone to TB due to an "overcrowded environment" might be an oversimplification. Consider elaborating on why students are more susceptible to TB, such as dense accommodations, classroom environments, or interaction with many individuals. In addition, it is reported that 306 students hadn't heard about TB, but it's not clarified if this means they scored zero in terms of TB's KAP. The authors should clarify if these 306 students were included in the final KAP assessment or excluded due to their lack of knowledge. Furthermore, the conclusion mentions that over a quarter of participants lacked any knowledge about TB, which doesn't entirely align with the previously stated 26.5% data. Please rectify this data for consistency. Lastly, the abstract lacks some key details about the research methods, such as the method of sample selection or the rationale for choosing specific data analysis techniques. The authors should expand on the research methods' description slightly if space allows.
Introduction Section:
The introduction provided offers a detailed context and background about Tuberculosis (TB), there are a few issues and areas of potential improvement:
· Consider simplifying the presentation of background information to ensure that the introduction is concise and directly relates to the aim of the study.
· Grouping or organizing the various studies mentioned according to their geographic location or focus can help improve clarity. It will make it easier for the reader to understand the global context of TB-related KAP.
· Some sentences are quite long and might benefit from being split into shorter, clearer statements.
· While the purpose of the study is mentioned towards the end of the introduction, consider making it more prominent or clearly distinguished from the rest of the background information.
· After discussing various studies, it would be helpful to have a transition sentence or two that succinctly and clearly leads into the purpose and significance of your own study.
· The paper should have specific and definitive research questions.
Methods Section:
· The description is redundant—Participants who are over 18 years of age and who are students at TU are included in the study. Participants who are less than 18 years of age, or who are from other universities were excluded from the study. It should be— The study included participants who were students at TU and over 18 years of age. Those less than 18 or from other universities were excluded.
· Where is the source of the formula used in the paragraph below the subheadings of 2.3 “Sample size calculation”?
· It seems that there are no missing values on the background variables of the study. However, there is no relevant description of other variables. The authors should clarify if it is the case. What is the rate and pattern of missing values? Is the pattern of missing values MCAR or MNAR?
· Are there control variables in the study?
· The authors should further clarify if there are incentives provided to the participants in the manuscript, if applicable.
· Below the subheadings “2.5” on page 3, the scores that the authors obtained are raw score. How do the authors justify that “agree =4” equals to “usually=4”? You do not have any reasons to do so unless you do something with them.
Results Section:
· The reviewer noticed that the decimals that appear in the manuscript with a trailing zero are inappropriate in format. Please note that many authorities in scientific, technical, and medical fields recommend that a zero should not be inserted before a decimal fraction when the number not be inserted before a decimal fraction when the number cannot be greater than 1 (e.g., correlations, proportions, and levels of statistical significance); that is, "p < 0.05" should be written as "p < .05."
· All tables are messy in the manuscript and please reformat them in an appropriate manner.
The authors need to improve English writing skills.
Author Response
Response to REVIEWER #2
Abstract Section:
Comment #1:
The description that students are prone to TB due to an "overcrowded environment" might be an oversimplification. Consider elaborating on why students are more susceptible to TB, such as dense accommodations, classroom environments, or interaction with many individuals. In addition, it is reported that 306 students hadn't heard about TB, but it's not clarified if this means they scored zero in terms of TB's KAP. The authors should clarify if these 306 students were included in the final KAP assessment or excluded due to their lack of knowledge. Furthermore, the conclusion mentions that over a quarter of participants lacked any knowledge about TB, which doesn't entirely align with the previously stated 26.5% data. Please rectify this data for consistency. Lastly, the abstract lacks some key details about the research methods, such as the method of sample selection or the rationale for choosing specific data analysis techniques. The authors should expand on the research methods' description slightly if space allows.
Response to Comment #1: We thank the reviewer for the comment. The statement about the students’ risk of getting TB infection was clarified. Also, the 306 students who did not hear about TB were not included in KAP analysis. This was clearly stated in the abstract, last paragraph of the introduction and the Results section: 3.1. Regarding the conclusion section, we used the term "a quarter" to indicate a near and close figure (a percentage of 26.5%), which are very close giving the same impression as a conclusion without repeating numbers of results. Since the abstract must be short and succinct, we tried to avoid providing many details about the methods and analytical technique. These issues were detailed in the Methods section. However, as suggested by the reviewer, we amended the sample collection method in the abstract to be slightly more detailed.
Introduction Section:
Comment #2: The introduction provided offers a detailed context and background about Tuberculosis (TB), there are a few issues and areas of potential improvement:
Response to Comment #2: We thank the reviewer for commending the introduction section.
Comment #3: Consider simplifying the presentation of background information to ensure that the introduction is concise and directly relates to the aim of the study.
Response to Comment #3: We thank the reviewer for this comment; and the manuscript was modified to reflect the reviewer’s suggestion. The introduction was revised, rearranged and shortened while presenting the related background knowledge of the research.
Comment #4: Grouping or organizing the various studies mentioned according to their geographic location or focus can help improve clarity. It will make it easier for the reader to understand the global context of TB-related KAP.
Response to Comment #4: The introduction was revised to reflect the reviewer’s suggestion. The study in Thailand was moved after that in Bangladesh (Asia), followed by the Cameroonian study; and we presented the two studies in Europe to be consecutive to each other. This paragraph was also rearranged and shortened.
Comment #5: Some sentences are quite long and might benefit from being split into shorter, clearer statements.
Response to Comment #5: We thank the reviewer for the comment. The revised version of the manuscript was modified to reflect the reviewer’s suggestion. Additionally, the manuscript was edited by a native English speaker.
Comment #6: While the purpose of the study is mentioned towards the end of the introduction, consider making it more prominent or clearly distinguished from the rest of the background information.
Response to Comment #6: Thanks. In this revised version, we clearly distinguished (italicized) the aim of the study at the end of the introduction. The aim was also stated in the abstract.
Comment #7: After discussing various studies, it would be helpful to have a transition sentence or two that succinctly and clearly leads into the purpose and significance of your own study.
Response to Comment #7: We thank the reviewer for the comment, and we have revised the manuscript while following the reviewer’s suggestion.
Comment #8: The paper should have specific and definitive research questions.
Response to Comment #8: The aim of this study was to assess the KAP levels among Taif University students regarding TB. This aim was clearly stated in the abstract as well as in the last paragraph of the introduction.
Methods Section:
Comment #9: The description is redundant—Participants who are over 18 years of age and who are students at TU are included in the study. Participants who are less than 18 years of age, or who are from other universities were excluded from the study. It should be— The study included participants who were students at TU and over 18 years of age. Those less than 18 or from other universities were excluded.
Response to Comment #9: We thank the reviewer for this comment. We followed the reviewer's suggestions while preparing the revised version of our manuscript.
Comment #10: Where is the source of the formula used in the paragraph below the subheadings of 2.3 “Sample size calculation”?
Response to Comment #10: The Equation is clearly stated in the third line of this section (n=72 x p (1-p)/e2).
Comment #11: It seems that there are no missing values on the background variables of the study. However, there is no relevant description of other variables. The authors should clarify if it is the case. What is the rate and pattern of missing values? Is the pattern of missing values MCAR or MNAR?
Response to Comment #11: That is true. The tool of this study was an online google form questionnaire. Our design of the questionnaire ensured that answering all questions is compulsory. For every question, we provided all the possible answers. If the student is not willing to take the survey, he could quit from the first question of the informed consent agreement. But, if the student decided to take the survey, and did not have any idea about what is T.B. then, choosing "I never heard about T.B." would be the terminating answer for his/her survey. Answering, "Yes, I heard about T.B." then he/she could answer the rest of the compulsory questions of the KAP survey regarding T.B.
Comment #12: Are there control variables in the study?
Response to Comment #12:
Only few questions were used as internal controls, at least one question in each of the main sections of the questionnaire (knowledge, attitude and practice) with the purpose of keeping internal consistency and ensuring clarity and understandability of the questions as well as confirming that different ways of asking would get the same answers.
Comment #13: The authors should further clarify if there are incentives provided to the participants in the manuscript, if applicable.
Response to Comment #13: There were no incentives for participation in this study. We mentioned this point in our revised manuscript after raising this question by the reviewer.
Comment #14: Below the subheadings “2.5” on page 3, the scores that the authors obtained are raw score. How do the authors justify that “agree =4” equals to “usually=4”? You do not have any reasons to do so unless you do something with them.
Response to Comment #14: For all the scores we prepared questions with answers that fits with the grading of Lickert scale, which is widely used in literature. These answers stand for keeping the middle answer as (equivocal/neutral or unclear) with having two levels above this middle with the most upper answer represents the extreme answer (maximum, very, always). The remaining two answers come below the middle one by showing the low/less state while keeping the least answer as Never/lowest. In the scoring of these answers usually Lickert scale gives 3 to the middle answer, 5 to the extreme upper answer and 1 to the extremely lower one. For each section of the Arabic version of the online questionnaire, we keep on using the same answers for all the 8 questions of practice. These details were presented to the Methods section.
Results Section:
Comment #15: The reviewer noticed that the decimals that appear in the manuscript with a trailing zero are inappropriate in format. Please note that many authorities in scientific, technical, and medical fields recommend that a zero should not be inserted before a decimal fraction when the number not be inserted before a decimal fraction when the number cannot be greater than 1 (e.g., correlations, proportions, and levels of statistical significance); that is, "p < 0.05" should be written as "p < .05."
Response to Comment #15: Regarding this formatting issue, we have fixed it in the current revised version of the manuscript according to the reviewer’s suggestion.
Comment #16: All tables are messy in the manuscript and please reformat them in an appropriate manner.
Response to Comment #16: All the tables were reformatted to fit the journal's style.
Comment #17: The authors need to improve English writing skills.
Response to Comment #17: The manuscript was edited by a native English speaker. An editing certificate is submitted along with the revision.
Round 2
Reviewer 2 Report
Review Comments
September 22, 2023
1. Abstract section could be revised in the following manner:
Abstract:
Tuberculosis (TB) remains a significant public health concern worldwide. Given the dense living and interactive nature of university environments, students may be at higher risk. This cross-sectional study assessed Tuberculosis-related Knowledge, Attitudes, and Practices (KAP) among students at Taif University (TU) from November 2022 to May 2023. Using a self-administered online questionnaire with 40 items, 1155 students participated. Key demographics: 68.2% females, 96.9% Saudi citizens, 94.5% unmarried, and 87.5% non-smokers. Of the respondents, 26.5% had no knowledge of TB. The TB-related KAP scores among the aware students were 64.9%, 74.8%, and 81% respectively. Medical college students exhibited significantly higher TB-related knowledge and attitudes than their non-medical peers (p < .001). The findings indicate a commendable level of TB-awareness among TU students, but there remains a substantial uninformed segment. Campaigns to enhance TB knowledge among TU students are suggested.
2.On page 2, paragraph 2 of the manuscript, the authors state: 'That study found that a deficiency exists in the general knowledge of non-medical students regarding TB [7].' This phrasing is ambiguous and imprecise. As a researcher, it's your responsibility to clearly highlight the gaps in the research, rather than glossing over them and leaving it to readers to search through your references.
3.Until now, the authors haven't provided clear research questions, which is inexcusable. Using common academic practices as a justification is unprofessional. Many of these practices in academia are misconceived. Research hypotheses or objectives can't substitute for well-defined research questions.
4.In Table 1, the four column names should be: Variable, subcategory, frequency, and Percent, respectively. The term 'college' should be revised to 'type of college’.
5.All tables should only display three lines: the first line, the second line, and the last line. All other lines should be hidden.
6. In previous feedback, the reviewer asked the authors to specify the source of their sample size calculation formula. Merely citing a software company isn't sufficient. Readers and reviewers seek clarity on the formula's scientific and accurate basis, not just the way it's presented by the authors.
7. In Table 1, all three columns should be left-aligned. Moreover, some variable names in the first column are questionable. For instance, 'social status' is inappropriate given that the content presented by the authors pertains to marital status. This cannot correspond. The authors should revise it to 'marital status'. Furthermore, in Table 1, the percentages in the fourth column should be right aligned at the decimal point.
8. What's particularly absurd is, didn't the authors review their own paper before submitting the manuscript? Why does the 'social status' section sum up to 94.5% + 5.2% + 3%, which exceeds 100%? Shouldn't the authors deeply reflect on how such a flawed submission was even made? Similarly, the same error occurs in the places of 'smoking' and 'Did you know who had the TB?' sections. It's concerning to see these discrepancies in the manuscript.
9. Please remove this sentence “Have you heard about Tuberculosis (TB)? N (%)” from Table 2.
10. In Table 2, there's a missing column for effect size. Relying solely on the p-value is insufficient; this oversight is significant.
11. Please revise Table 3 following the feedback provided for Table 1.
12. Please revise Table 4, 5,6,7, and 8 following the feedback provided for Table 2.
The authors should work hard improving English writing skills.
Author Response
Response to "REVIEWER #2"
COMMENT #1. Abstract section could be revised in the following manner:
Abstract:
Tuberculosis (TB) remains a significant public health concern worldwide. Given the dense living and interactive nature of university environments, students may be at higher risk. This cross-sectional study assessed Tuberculosis-related Knowledge, Attitudes, and Practices (KAP) among students at Taif University (TU) from November 2022 to May 2023. Using a self-administered online questionnaire with 40 items, 1155 students participated. Key demographics: 68.2% females, 96.9% Saudi citizens, 94.5% unmarried, and 87.5% non-smokers. Of the respondents, 26.5% had no knowledge of TB. The TB-related KAP scores among the aware students were 64.9%, 74.8%, and 81% respectively. Medical college students exhibited significantly higher TB-related knowledge and attitudes than their non-medical peers (p < .001). The findings indicate a commendable level of TB-awareness among TU students, but there remains a substantial uninformed segment. Campaigns to enhance TB knowledge among TU students are suggested.
Response to Comment #1: Thank you very much. We really appreciate this help, and we used the suggested Abstract in the manuscript.
COMMENT #2. On page 2, paragraph 2 of the manuscript, the authors state: 'That study found that a deficiency exists in the general knowledge of non-medical students regarding TB [7].' This phrasing is ambiguous and imprecise. As a researcher, it's your responsibility to clearly highlight the gaps in the research, rather than glossing over them and leaving it to readers to search through your references.
Response to Comment #2: The sentence was re-written to be clearer to the readers.
COMMENT #3. Until now, the authors haven't provided clear research questions, which is inexcusable. Using common academic practices as a justification is unprofessional. Many of these practices in academia are misconceived. Research hypotheses or objectives can't substitute for well-defined research questions.
Response to Comment #3: The last paragraph of the introduction was amended to explain why this research was performed.
COMMENT#4. In Table 1, the four column names should be: Variable, subcategory, frequency, and Percent, respectively. The term 'college' should be revised to 'type of college’.
Response to Comment #4: We corrected Table 1 following the reviewer’s suggestions by making four columns (Variable, subcategory, frequency and percent) and correcting the "type of college".
COMMENT #5. All tables should only display three lines: the first line, the second line, and the last line. All other lines should be hidden.
Response to Comment #5: This comment was followed for all tables.
COMMENT #6. In previous feedback, the reviewer asked the authors to specify the source of their sample size calculation formula. Merely citing a software company isn't sufficient. Readers and reviewers seek clarity on the formula's scientific and accurate basis, not just the way it's presented by the authors.
Response to Comment #6: A Raosoft Inc., (Seattle, WA) calculator was used to determine the sample size using the Equation n=72 x p (1-p)/e2 for which n is the sample size, z (1.96) is the z-score associated with a level of confidence (95%), p is the sample proportion and is expressed as a decimal, and e (.05) is the margin of error expressed as a decimal. The calculated sample size was at least 377 participants. A reference was added to this method in the revised manuscript.
COMMENT #7. In Table 1, all three columns should be left-aligned. Moreover, some variable names in the first column are questionable. For instance, 'social status' is inappropriate given that the content presented by the authors pertains to marital status. This cannot correspond. The authors should revise it to 'marital status'. Furthermore, in Table 1, the percentages in the fourth column should be right aligned at the decimal point.
Response to Comment #7: Thanks. The comment was followed regarding the design of the table and changing the social into marital status.
COMMENT #8. What's particularly absurd is, didn't the authors review their own paper before submitting the manuscript? Why does the 'social status' section sum up to 94.5% + 5.2% + 3%, which exceeds 100%? Shouldn't the authors deeply reflect on how such a flawed submission was even made? Similarly, the same error occurs in the places of 'smoking' and 'Did you know who had the TB?' sections. It's concerning to see these discrepancies in the manuscript.
Response to Comment #8: Thanks a lot for checking these numbers and percentages. This mistake appeared when we changed (0.3 into .3), where the (.) was mistakenly deleted. The percentages are 94.5 + 5.2 + .3 = 100%.
COMMENT #9. Please remove this sentence “Have you heard about Tuberculosis (TB)? N (%)” from Table 2.
Response to Comment #9: The sentence was removed according to this comment.
COMMENT #10. In Table 2, there's a missing column for effect size. Relying solely on the p-value is insufficient; this oversight is significant.
Response to Comment #10: A new column was inserted for this table that contains Odds ratio and confidence interval.
COMMENT #11. Please revise Table 3 following the feedback provided for Table 1.
Response to Comment #11: Table 3 was revised as requested.
COMMENT #12. Please revise Table 4, 5,6,7, and 8 following the feedback provided for Table 2.
Response to Comment #12: Tables 4,5,6,7,and 8 were revised as requested.
